# Recent Advances in Peptide Inhibitors Targeting Wild-Type Ras Protein Interactions in Cancer Therapy

**DOI:** 10.3390/ijms26041425

**Published:** 2025-02-08

**Authors:** Weirong Qin, Zijian Liu, Mingyu Huang, Lin Liang, Yuxin Gan, Zubei Huang, Jin Huang, Xiangzan Wei

**Affiliations:** 1Pharmaceutical College, Guangxi Medical University, Nanning 530021, China; m13316640542@163.com (Z.L.); m17377084344@163.com (M.H.); leunglum@163.com (L.L.); 18878748220@139.com (Y.G.); 13977465949@163.com (Z.H.); huangjin@mailbox.gxnu.edu.cn (J.H.); 2Guangxi Key Laboratory of Bioactive Molecules Research and Evaluation, Guangxi Medical University, Nanning 530021, China; 3Key Laboratory of Biological Molecular Medicine Research (Guangxi Medical University), Education Department of Guangxi Zhuang Autonomous Region, Nanning 530021, China; 4State Key Laboratory of Chemical Oncogenomics, Guangdong Provincial Key Laboratory of Chemical Genomics, Peking University Shenzhen Graduate School, Shenzhen 518055, China

**Keywords:** Ras proteins, peptide inhibitors, cancer therapy, protein–protein interaction

## Abstract

Ras proteins are pivotal in the regulation of cell proliferation signals, and their dysregulation is intricately linked to the pathogenesis of various malignancies. Peptide inhibitors hold distinct advantages in targeting Ras proteins, attributable to their extensive binding domains, which result from the smooth surfaces of the proteins. The array of specific strategies includes the employment of full hydrocarbon chains, cyclic peptides, linear peptides, and N-terminal nucleation polypeptides. These methods effectively suppress the Ras signaling pathway through distinct mechanisms, highlighting their potential as anti-neoplastic agents. Moreover, cutting-edge methodologies, including the N-terminal aspartate nucleation strategy and the utilization of hydrocarbon-stapled peptides, are transforming the landscape of therapeutics aimed at Ras proteins. These innovations highlight the promise of peptide libraries and combinatorial chemistry in augmenting binding affinity, specificity, and cellular permeability, which are pivotal for the development of potent anti-cancer agents. The incorporation of dual therapeutic strategies, such as the synergy between peptide inhibitors and conventional chemotherapy or the use of radiotherapy enhancers, emerges as a compelling strategy to bolster the efficacy of cancer treatments targeting the Ras-MAPK pathway. Furthermore, recent studies have demonstrated that Ras-targeting stabilized peptides can amplify the radio-sensitivity of cancer cells, offering an innovative approach to enhance the efficacy of radiation therapy within cancer management.

## 1. Introduction

Cancer is a predominant cause of mortality on a global scale, with radiotherapy (RT) being one of the three principal therapeutic approaches in oncology [1]. Despite the relentless progress in radiotherapy technology, the radio-resistance of tumor cells poses a substantial challenge to the effectiveness of treatment [2]. Consequently, enhancing the radiosensitivity of tumor cells has emerged as a pivotal research priority in contemporary cancer research [3]. Among the myriad of pathways that modulate tumor radiation sensitivity, the Ras signaling pathway has attracted substantial interest due to its pivotal role in cellular proliferation, differentiation, and survival.

The Ras protein family is instrumental in the regulation of mitotic signaling and is intimately connected with cellular processes, including cell proliferation [4]. Literature reviews indicate that Ras oncogenes are frequently deregulated in human malignancies, with the hyperactivation of Ras/MAPK signaling pathways potentially contributing to the genesis and progression of numerous tumors [5,6,7,8]. Therefore, inhibiting the Ras signaling pathway is regarded as a promising therapeutic strategy to augment the radio sensitivity of tumor cells [9].

## 2. Ras Signaling Pathways

Ras proteins are situated at the apex of the Ras-Raf-MEK-ERK signaling pathway, with their activation being contingent upon nucleotide GTP exchange, a process facilitated by Sos. The development of stable α-helical polypeptide inhibitors that disrupt the Ras-Sos interaction could potentially prevent the propagation of downstream signaling pathways activated by Ras [9,10]. Recent research trends in the realm of Ras oncoproteins are predominantly focused on several strategies: inhibiting the Ras-Sos interaction to impede downstream signaling [11,12,13,14], preventing the membrane localization of Ras on cellular membranes [15,16], blocking Ras-GTP nucleotide exchange to maintain the Ras protein in the inactive Ras-GDP state [17,18,19], and impeding the interaction of Ras proteins with downstream effector proteins such as Raf (Figure 1) [8,20,21]. Furthermore, the inhibition of phosphosignaling cascades among Raf, MEK, and ERK can also serve to suppress the Ras/MAPK signaling pathways [22,23].

Nevertheless, RAS exhibits a high affinity for GTP, characterized by a K_D_ in the picomolar scale [24], which poses a significant challenge for the development of competitive GTP analogs [25]. Certain small molecule compounds, including sulindac sulfide, DCAI, Kobe0065, compound 3144, and Rigosertib, have demonstrated the ability to inhibit the interaction between Ras proteins and their effector proteins [26,27]. While targeting protein–protein interactions is effective and promising in drug development, the creation of small molecule modulators is exceptionally challenging due to the smooth and flat surface of wild-type Ras proteins, which lack the deep and hydrophobic binding pockets required for effective small molecule engagement [28,29,30,31]. Conversely, polypeptides, with their extensive surface area, offer significant advantages in interacting with the shallow and flat surfaces of Ras proteins [8,32]. Moreover, recent advancements in the development of cyclic peptides, linear peptides, and hydrocarbon-stapled peptides have improved binding affinity and selectivity for Ras proteins, highlighting their potential as potent cancer therapeutics (Table 1).

### 2.1. Hydrocarbon Peptide

In 1998, Blackwell et al. pioneered the development of a ruthenium-catalyzed ring-closing olefin Metathesis (Ring Closing Metathesis: RCM) technique. This method utilizes O-allylserine residues located on adjacent helix turns, allowing for ring closure via olefin metathesis to construct stable cyclic polypeptides for broader biological applications (Figure 2A) [39]. Building upon this foundation, Schafmeister et al. and Walensky et al. harnessed the olefin metathesis reaction, as initially developed by Blackwell et al. to cyclize amino acid side chain residues and synthesize all-hydrocarbon side chains. This advancement led to the synthesis of all-hydrocarbon side chains, which significantly improved the helicity, stability, protease resistance, and cell membrane permeability of the polypeptides (Figure 2B) [40,41]. Furthermore, the olefin metathesis ring-closing reaction allows for the cyclization of amino acid side chain residues at the (i, i + 4) or (i, i + 7) positions on the polypeptide backbone, thereby facilitating the construction of a stable α-helical polypeptide structure [41].

In 2011, Patgiri et al. engineered α-helical polypeptides, designated as **HBS 3** (Table 1) [11], which specifically targeted to HRas using the hydrogen bond surrogate (HBS) method [42]. The **HBS 3** peptide, with the sequence XFE*GIYRLELLKAEEAN-NH_2_, has been shown to inhibit Ras activation mediated by Sos, subsequently suppressing downstream signal transduction. This approach to peptide design has been critical in enhancing the stability and functionality of peptides in their interaction with Ras proteins. The innovative use of the **HBS** method in the development of **HBS 3** underscores its potential in the field of cancer therapeutics, offering a promising direction for future research and treatment strategies. In 2015, Leshchiner et al. introduced an all-hydrocarbon “stapled peptide” known as **SAH-SOS1_A_** (Table 1) for both wild-type and mutant KRAS (Figure 3A). This peptide notably reduced the phosphorylation levels of ERK and AKT within the downstream Ras signaling cascade, effectively curbing the proliferation of cancer cells carrying mutated genes (Figure 3B,C) [12]. In 2023, Li et al. synthesized a series of hydrocarbon-stapled peptides that mimicked the α-helical conformation of SOS1, thereby dampening pan-Ras activity. Among these peptides, **SSOSH-5** (Table 1) stood out with its high affinity for HRas and its potent ability to induce cell apoptosis [10].

### 2.2. Cyclization Peptides

In 2013, Wu et al. screened a compound known as cyclic peptide of **compound 12** (Table 1), which binds to K-Ras with submicromolar affinity within a large combinatorial library of K-Ras targeted cyclic peptides. This peptide exhibited submicromolar affinity for K-Ras, demonstrating its potential to disrupt the critical interaction between K-Ras and its effector proteins, including Raf, Ral-GDS, and Tiam1 [33], which could significantly affect cancer cells. In 2015, Upadhyaya et al. engineered a bicyclic peptide, **cyclorasin 9A5** (Table 1), which showcased remarkable cell-penetrating capabilities. This peptide was designed to inhibit Ras-effector interactions by binding to Ras-GTP, thereby triggering apoptosis in cancer cells (Figure 4) [34]. In 2016, Trinh et al. synthesized a series of permeable bicyclic peptides by strategically fusing cyclic peptides with ringed cell-penetrating peptides. Through rigorous screening and optimization processes, they successfully identified effective K-Ras inhibitor **peptides 49** and **peptides 54** (Table 1), which were capable of inducing apoptosis in cancer cells [35]. In 2020, Zhang Z et al. identified a cyclic peptide ligand, **KD2** (Table 1), that selectively binds to KRas in a highly dynamic pocket near the switch II region (SIIP). This interaction results in a significant displacement of the α2 helix and the Switch II loop, effectively blocking Ras-Raf interactions and inhibiting the proliferation of cancer cells [36].

### 2.3. Linear Peptides

In 1996, Schott M E et al. conducted a study on linear peptides capable of inducing T helper site mutations in RAS immunogens. Their preliminary investigations, which involved the use of point-mutated linear peptide epitopes, revealed that the most robust in vitro proliferation response was elicited following the subcutaneous administration of peptides in adjuvant formulations enriched with squalene [43]. Typically, linear peptides are designed to target the switch regions of RAS proteins, which play a pivotal role in the activation and deactivation cycles of these proteins. By binding to these regions, peptides have the potential to suppress RAS activity and the subsequent signaling cascades, thereby potentially yielding anti-cancer effects.

### 2.4. Metallopeptides

In 2022, Learte-Aymamí S et al. discovered a short α-helical peptide, **αH-His_2_** (Table 1), capable of chelating with platinum. This interaction significantly boosts the α-helix content of **αH-His_2_**, thereby enhancing its conformational stability and responsiveness to stimuli [37]. Experimental observations indicated that the metallopeptide exhibits a more stable fold and a pronounced propensity to adopt alpha-helix configurations. The folded conformation of these metallopeptides facilitates their interaction with cellular components, enabling efficient internalization into living cells.

Furthermore, it has been demonstrated that the metallopeptide **αH-His_2_** [Pd] binds to KRAS, which is a critical step in inhibiting the MAPK kinase cascade and ultimately leading to the suppression of cancer cell proliferation. The binding of **αH-His_2_** [Pd] to KRAS was confirmed by fluorescence anisotropy, which showed a marked increase in anisotropy upon the formation of the palladium/peptide complex, consistent with binding to KRAS_wt_ and the formation of a larger complex with slower rotational diffusion [37]. The addition of a Pd (II) chelator, DEDTC, to the solution of the complex **αH-His_2_** [Pd]/KRAS_wt_ resulted in a marked decrease in the TMR anisotropy, indicating the disassembly of the metallopeptide **αH-His_2_** [Pd], and disengagement of **αH-His_2_** from KRAS_wt_. This confirms the direct interaction of **αH-His_2_** with KRAS and the reversibility of the system.

The functional consequences of the formation of the **αH-His_2_** [Pd]/KRAS complex include the modulation of KRAS’s GTPase activity, as demonstrated by the inhibition of the nucleotide release process. Additionally, the preformed metallopeptide **αH-His_2_** [Pd] was internalized by A549 cells, and the peptide inhibited the MAPK RAF-MEK-ERK cascade in a dose-responsive manner, as shown by the decrease in the levels of phosphorylated ERK.

### 2.5. N-Cap Helix Nucleation Peptides

In 2016, Zhao et al. employed 2,3-diaminopropionic acid (Dap) and the N-terminal of peptides to form an amide bond with L-aspartic acid at the (i, i + 3) positions, thereby devising an N-terminal aspartic acid nucleation strategy (TD strategy). This innovative approach stabilizes peptides into an α-helical structure (Figure 5), resulting in improved stability and cell membrane penetration. These findings were reported in the prominent journal “Angewandte Chemie” [44]. The TD strategy was successful across various biological targets, culminating in the development of peptide inhibitors for the estrogen receptor ER α [45], the histone deacetylase HDC1 protein [46], and stable peptides inhibitors [47] for USP 30. Collectively, these studies have demonstrated that stable peptides constructed using this strategy exhibit high helicity, excellent serum stability, and potent cell membrane penetration capabilities.

In 2024, Qin et al. extended the scope of the N-terminal aspartate nucleation strategy in biological applications by designing and synthesizing (i, i + 3) α-peptide inhibitors tailored to the crystal structure of the Ras-Sos complex (Figure 6) [9]. Concurrently, they utilized a whole hydrocarbon strategy to synthesize the control peptide **HBS 3** [11]. In contrast to the whole hydrocarbon strategy, which necessitates an 80 °C oil bath and is labor-intensive, the “N-terminal aspartate nucleation strategy” offers a more straightforward approach. It operates under milder conditions, facilitating reactions at room temperature under catalytic influence, without the incorporation of non-natural amino acid side chains. Subsequent experiments revealed that the peptides **H5** and **H2** (Table 1) exhibit a high binding affinity for HRas, enabling them to selectively target and eliminate cancer cell lines with elevated Ras expression. Peptides **H5** and **H2** (Table 1) can effectively inhibit the Ras/MAPK signaling pathway and down regulate the phosphorylation levels of downstream kinases such as ERK. Additionally, they have been shown to significantly enhance the radiation sensitivity of cervical cancer cells [9].

### 2.6. Other Peptides

In 2021, Arora’s team successfully engineered a structurally stable Sos protein mimic, designated as **CHD^Sos^-5** (Table 1), which non-covalently interacts with wild-type and oncogenic Ras proteins, modulating the downstream kinase signaling pathways. This protein mimic exhibits resistance to proteolysis and has the ability to penetrate cancer cells, selectively exerting toxicity through macropinocytosis, particularly in cells harboring oncogenic Ras mutations. This discovery lays a theoretical foundation for the development of non-covalent, targeted inhibitors for Ras mutant-driven cancers [38].

## 3. Conclusions

The exploration of Ras signaling inhibition through peptide inhibitors is a burgeoning area of research that enhances our understanding of the molecular mechanisms underlying Ras-driven oncogenesis while opening up new avenues for therapeutic intervention across various cancer types. Ongoing efforts to refine peptide structures, improve their permeability across cellular membranes, and integrate them into existing treatment protocols are expected to broaden the scope of targeted therapies against Ras proteins. The translation of these findings into clinical practice will be pivotal for establishing peptide inhibitors within oncological treatment paradigms. This approach highlights the potential of peptide-based modulation to contribute significantly to the fight against cancer, suggesting a promising outlook for managing Ras-related malignancies.

The Ras protein is a critical regulator of cell proliferation and differentiation, and its dysregulation is frequently associated with the development and progression of various cancer treatments [48]. Advancements in peptide design and synthesis, including the N-terminal aspartic acid nucleation strategy, have resulted in stable α-helix polypeptides with enhanced stability and cell membrane permeability compared to their linear counterparts [49]. This is particularly significant for the therapeutic potential of peptides, as it improves their ability to reach intracellular targets.

Recent advancements in biochemical and structural biology, coupled with computational biology insights, have provided a deeper understanding of the structure and dynamics of Ras proteins [48]. This knowledge is pivotal for identifying new peptide candidates and binding sites, leading to the development of more potent inhibitors. For instance, peptides that target the Ras-GTP state have been identified, which could enhance inhibitory effects by specifically interacting with the active conformation of Ras.

In summary, the growing understanding of the Ras signaling pathway, alongside technological advancements in peptide stability, positions RAS-targeted stable peptides as key drug candidates in oncological therapy. These developments offer novel strategies and a beacon of hope for enhancing therapeutic outcomes and bolstering patient survival rates. Future investigative efforts must focus on harnessing the clinical translational potential of polypeptide drugs, refining their efficacy in practical settings through pioneering methods, and thereby facilitating broader and more impactful cancer treatment.

## Figures and Tables

**Figure 1 ijms-26-01425-f001:**
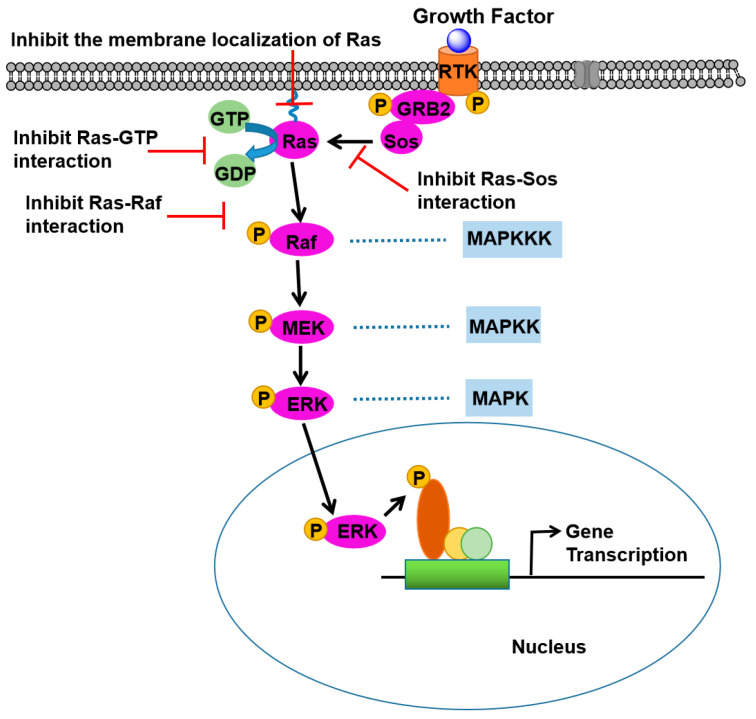
Several common strategies have been developed to target Ras proteins in order to block the Ras/MAPK signaling pathway. Solid black arrows indicate direct activation, dotted blue lines represent kinase cascades, red arrows with bars denote inhibitory actions targeting specific interactions, and the blue arrow shows the translocation of an activated protein to the nucleus to regulate gene transcription. These strategies include the inhibition of the Ras-SOS interaction, prevention of Ras membrane localization, disruption of the RAS-GTP interaction, and inhibition of the Ras-Raf interaction.

**Figure 2 ijms-26-01425-f002:**
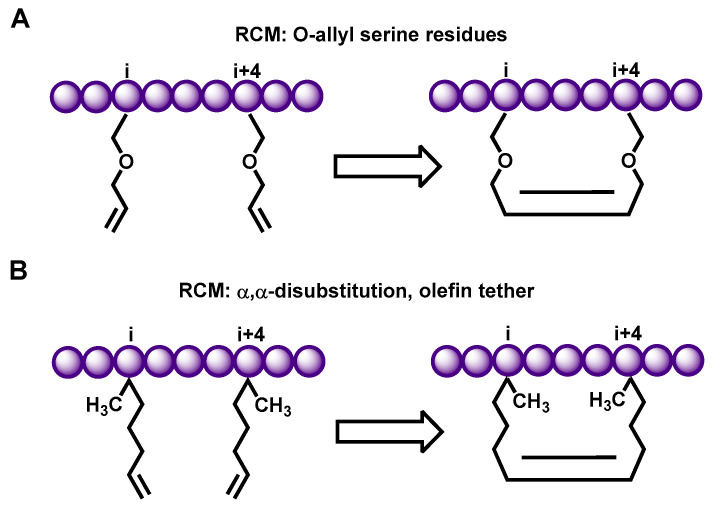
The closure of the amino acid side chains of the polypeptide was accomplished using ruthenium-catalyzed ring-closing olefin Metathesis (RCM) [39,40,41]. (**A**) Blackwell et al. developed a strategy for the olefin metathesis reaction on two o-allyl amino acid residues. (**B**) Schafmeister et al. and Walensky et al. utilized α, α-disubstituted unnatural amino acids with all hydrocarbon chains to facilitate ring closure, resulting in stable peptides with a significant α-helix structure.

**Figure 3 ijms-26-01425-f003:**
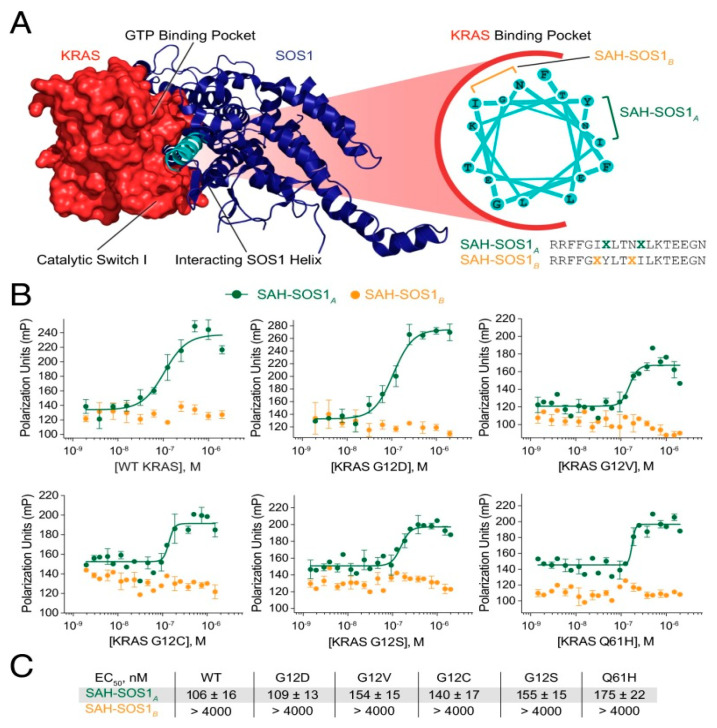
Design and KRAS binding activity of SAH-SOS1 peptides. (**A**) The crystal structure of the KRAS-SOS1 protein complex is depicted alongside the “staple peptide” **SAH-SOS1A**. The loop closure between two X amino acids, where X represents a 4-pentenoic acid residue, in the polypeptide sequence was achieved using a full hydrocarbon stapling strategy. (**B**) Fluorescence polarization binding analysis was performed on FITC-labeled SAH-SOS1 peptide and recombinant KRAS protein, which included both wild-type and mutant variants. (**C**) Table of EC50 values for binding interactions between SAH-SOS1 peptide and single KRAS protein. (Reprinted with permission from Ref. [12]. Copyright (2015) Proceedings of the National Academy of Sciences of the United States of America).

**Figure 4 ijms-26-01425-f004:**
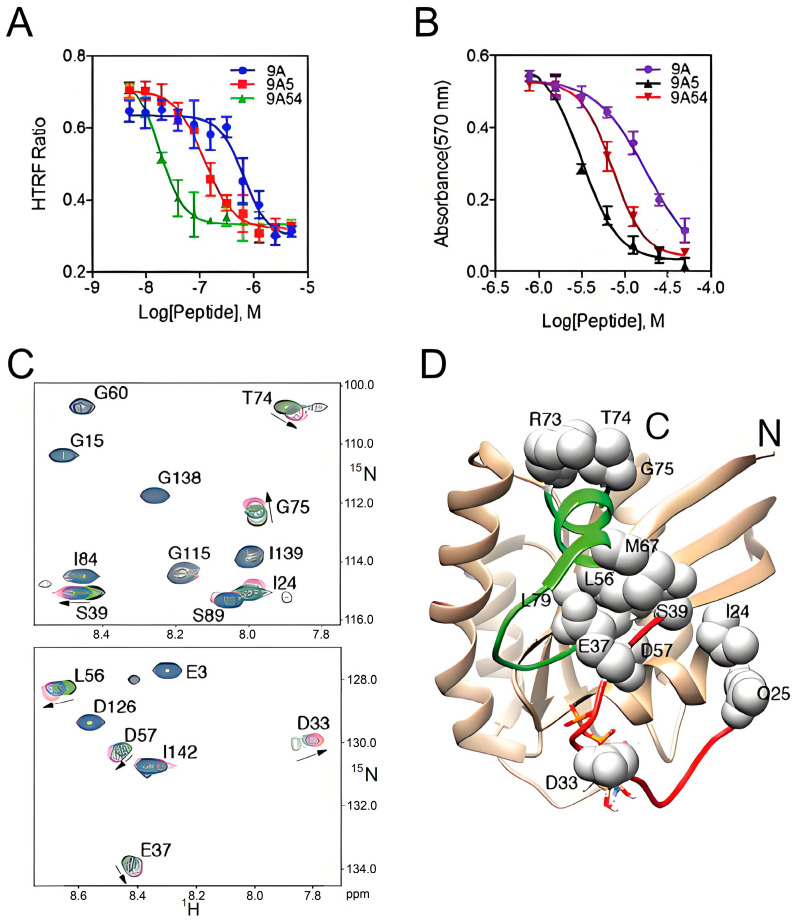
K-Ras binding assay, cellular uptake, and the anti-proliferative activity of selected cyclic peptides. (**A**) The inhibitory effect of 9A, 9A5, and 9A54 on Ras-Raf interaction, as measured by HTRF (Homogeneous Time-Resolved Fluorescence). (**B**) Effect of 9A, 9A5, and 9A54 on the cell viability of H1299 (a human lung cancer cell line). (**C**,**D**) 1H-15 N HSQC spectral superposition of K-RAS (G12V)-GDP in the presence of 0 (black), 0.5 equivalent (green), 0.75 equivalent (blue), and excess (red) 9A5 at 298 K. Panel D specifically shows the structure of wild-type K-RAS-GDP (PDB code 4LPK) with K-Ras residues interacting with 9A5 mapped onto it and displayed as spheres. Switch I and II regions are highlighted in red and green, respectively, and GDP is depicted in a stick shape representation. (Reprinted with permission from Ref. [34]. Copyright (2015) Angew. Chem. Int. Ed. Engl.).

**Figure 5 ijms-26-01425-f005:**
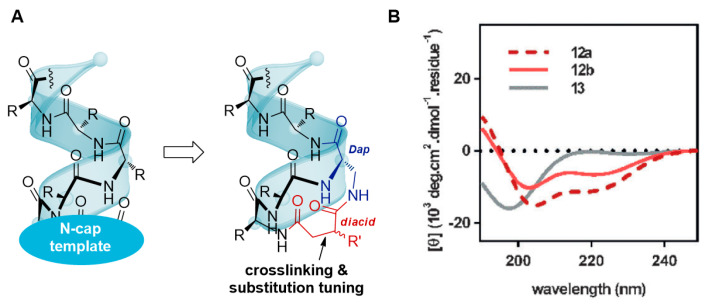
Schematic diagram of terminal aspartic acid nucleation (TD strategy) template. (**A**) Schematic representation of the N-terminal aspartate-nucleated template formed by the cross-linking of the Dap (aspartic acid) and isoAsp (isomer of aspartic acid) amino acids. This strategy involves the use of a diacid to cross-link the N-cap (N-terminal capping group) with the Dap, which is a key step in the formation of a stable α-helix. (**B**) Circular dichroism (CD) spectra of peptides 12a, 12b, and 13 in PBS (pH 7.4). Peptide 12a is the template-assembled peptide with a D-Asp to L-isoAsp cross-link at the N-terminus, which is designed to nucleate and stabilize an α-helical structure. Peptide 12b serves as a control with a D-Asp to D-Asp cross-link, and peptide 13 is the linear peptide without any cross-link, serving as a baseline to compare the effects of the TD strategy on peptide conformation. The CD spectra shows that peptides 12a and 12b exhibit characteristic α-helical conformations, as indicated by the negative band around 222 nm and a positive band around 208 nm, which are absent in the unmodified peptide 13, displaying a random coil conformation. This demonstrates the effectiveness of the TD strategy in promoting α-helical structure formation, which is essential for the biological activity of many peptides. (Reprinted with permission from Ref. [44]. Copyright (2016) Angewandte Chemie International Edition).

**Figure 6 ijms-26-01425-f006:**
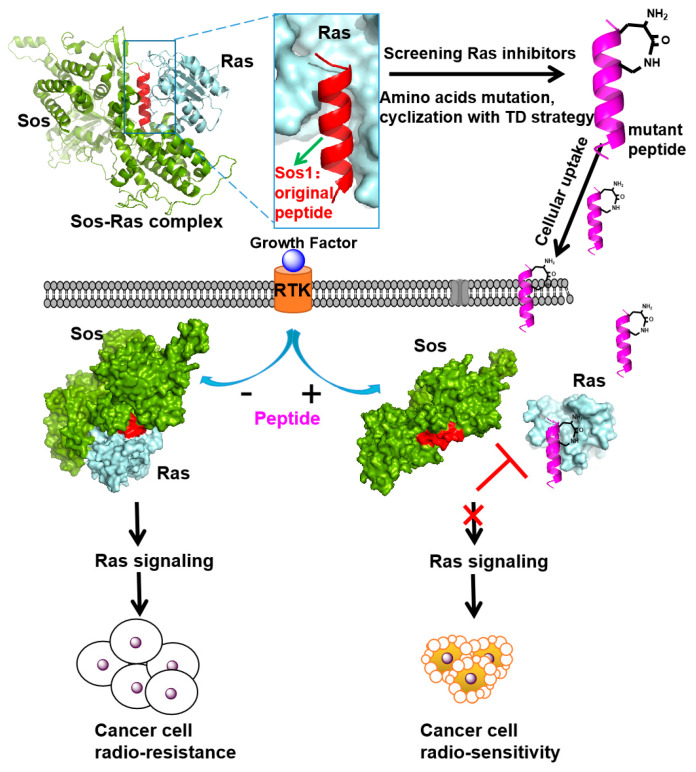
Schematic illustration of Sos1-based Ras-interacting helical peptides for increasing radio-sensitivity. RTK, receptor tyrosine kinase. The Sos1 original peptide represents the Sos1 α-helix (Amino acids 929–944: FFGIYLTNILKTEEGN), depicted in red. The mutant peptide, resulting from amino acids mutation and cyclization with the TD strategy, is shown in purple. Cyan indicates the Ras protein. Black arrows represent the direction of signaling, while the plus and minus signs indicate activation and inhibition, respectively. The red arrows with bars denote inhibitory actions targeting specific interactions, red Cross represents the blocking effect. (Reprinted with permission from Ref. [9]. Copyright (2024): Bioconjugate Chem).

**Table 1 ijms-26-01425-t001:** Sequences and binding affinities of the Ras peptide inhibitors.

Name	Peptide Sequence	Binding Affinity KRAS (μM)	Binding Affinity NRAS (μM)	Binding Affinity HRAS (μM)	Reference
**HBS 3**	XFE*GIYRLELLKAEEAN-NH_2_	N.D	N.D	28 ± 4.5	[11]
**SAH-SOS1_A_**	RRFFGI{Aaa}LTN{Aaa}LKTEEGN (Covalent bridge:Aaa7-Aaa11)	5–15	N.D	N.D	[12]
**SSOSH-5**	Ac-WIGRLLTS_5_IR^H^RS_5_RNGN-NH_2_	N.D	N.D	3.92	[10]
compound **12**	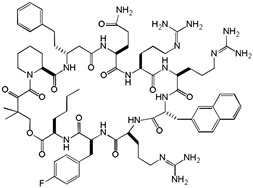	0.70	N.D	N.D	[33]
**9A5**	Cyclo({d-Ala}-RRRARF-{Nal}-QWT)	0.12	N.D	N.D	[34]
**bicyclic peptides** **(peptides 49)**	A-L-F-R-N-Pra*-I-D	0.21 ± 0.10	N.D	N.D	[35]
**bicyclic peptides** **(peptides 54)**	A-L-F-R-N-Pra-I-D	17 ± 11	N.D	N.D	[35]
**KD2**	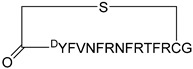	12.4	N.D	N.D	[36]
**αh-His_2_**	TMR-Ahx-RRFFGIHLTNHLKTEEGN	0.345	N.D	N.D	[37]
**H2**	H-WRR-cyclo(isoDFFDap)-IYLTNILKTEEGN-NH_2_	N.D	N.D	0.53	[9]
**H5**	H-WRR-cyclo(isoDFFDap)-IYLTNILKTQEGN-NH_2_	N.D	N.D	0.13	[9]
**CHD^Sos^-5**	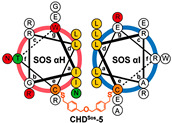	1.9 ± 0.3	N.D	N.D	[38]

(Abbreviations: Dap, 2,3-diaminopropionic acid; isoD, L-isoaspartic acid; X represents a 4-pentenoic acid residue; * = N-allyl residue; *G = N-allylglycine; N.D, not determined; Pra* = DCAI-modified propargylglycine).

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
