# Peer review of "Recent Advances in Peptide Inhibitors Targeting Wild-Type Ras Protein Interactions in Cancer Therapy"

_ijms, 2025, doi:10.3390/ijms26041425_

Round 1
Reviewer 1 Report
Comments and Suggestions for Authors
This is a well written and up-to-date review on the topic; I recommend publication with minor corrections; there are a few type errors, in one figure the resolution for better readability could be enhanced and the organisation of the entire text with headings and subheadings could be adapted and improved. Please find these suggestions detailed in the attached Word file. Thank you again, it was a pleasure reading your manuscript.

Author Response
Re: Manuscript Resubmission
Dear reviewer 1:
On behalf of my coauthors, I would like to re-submit the revised manuscript entitled “Recent advances in peptide inhibitors targeting wild-type Ras protein interactions in cancer therapy” [ijms-3387419] by Qin et al. for your evaluation.
In the past days, we carefully revised the manuscript based on the comments from the editor and all reviewers. We performed the suggested analyses including increasing the citations of references, improving the resolution of images, carefully checking and revising the sentences, introducing subheadings, etc.
We sincerely thank all of you for your in-depth comments/suggestions/questions and careful reading, which helped us tremendously in improving our manuscript. The attachment is the point-to-point answers to the editor’s and reviewers’ comments, and we hope our revisions could satisfy the rigorous requirements of all reviewers and the journal.
Thank you very much for reviewing our manuscript.
Cordially,
Weirong Qin, Ph.D.
Pharmaceutical College,
Guangxi Medical University,
Nanning, China 530021
- mail: qinweirong@sr.gxmu.edu.cn

Reviewer 2 Report
Comments and Suggestions for Authors
In their paper, Qina et al review the development of peptides as suppressors for the signaling pathway of Ras protein and their use in cancer therapy.
Overall, the manuscript is poorly written and organized. Being a review, the quality of citations should be high, but in this case, the references cited in this review are numerically limited (less than those expected for a review in this field) and sometimes inappropriately cited throughout the text. Some references appear to be outdated. Specific concerns are listed below.
Page 1, lines 33-37: please support the statements about radiotherapy with recent references.
Page 2, lines 44-47: Although the role of Ras in human malignancies is well known, I suggest to support these statements with some more literature, and more recent than the unique review cited (dated 2005, i.e. almost 20 years ago). Moreover, the statement “inhibiting the Ras signaling pathway is regarded as a promising therapeutic strategy to sugment the radiosensitivity of tumor cells” must be substantiated by citing consistent and updated literature evidence.
Same page, lines 50-51: the sentence “The development of stable alpha-helical polypeptide inhibitors [...] downstream signaling pathways activated by Ras” must be supported by literature citation, especially referring to the need of alpha-helical polypeptides for interaction with Ras.
Same page, lines 52-59: the references cited as examples of “recent research trends in the realm small molecule or polypeptide inhibitors targeting Ras oncogenic proteins” are generally quite old (published between 10 and 15 years ago). To substantiate this claim, authors need to find more recent work (published no more than 5 years ago).
Same page, lines 73-74: please substantiate the claim with references
Page 3, Table 1: please indicate the sources of the data included in the table. Moreover, Table 1 is not cited throughout the text, therefore it is not immediately clear to the reader that the data indicated in the table derive from the references discussed in the following paragraphs with respect to the table itself.
Same page, lines 83-90: this paragraph is inappropriate in this position, it sounds like a (premature) conclusion of the review.
Page 4, line 96 and legend to figure 2: the way to cite “Schafmeister and Walensky et al.” is inappropriate. Please cite “Schafmeister et al. and Walensky et al.”
Same page, line 112: please explain or cite the direct reference to the “hydrogen bond surrogate (HBS) method”
Same page, line 118: the citation to reference 23 in the text is incorrect. Either use “In 2023, Li et al...” or “In 2023, Zou and coworkers...”
Page 5, legend of figure 3. At line 124-125: please clarify the meaning of “two XX amino acids”: At line 128: please correct “EC 50” with “EC50” in the legend of figure 3
Same page, lines 143-147: including reference 27 in the list of “cyclization peptides” is inappropriate, since the cyclic RGD peptides are not involved in the active interaction with Ras proteins, but they target integrins to enhance the delivery of siRNA into tumor cells.
Same page, lines 151 and following: the Sos proteomimetic designed by Arora’s team is not a cyclic peptide, therefore its citation in this paragraph is inappropriate.
Page 6, legend to figure 4: please correct “...and antiproliferative activity of selected cyclic peptides” and explain the meaning of the acronyms (HTRF, H1299). Please specify that 4LPK is the PDB code of the structure represented in figure 4 (it would be useful to identify this as panel D)
Same page, paragraph 5: from the description reported in this paragraph, it is not at all evident that the alphaH-His2 peptide interacts directly with Kras.
Page 7, Figure 5: please explain what are peptides 12a, 12b and 13.
Page 8-9: the entire paragraph (lines 224 to 284) appears as a confusing recapitulation of things already discussed in the review, with generic sentences not substantiated by references. Moreover, lines 247-249 are a duplicate of lines 227-229
Finally, I found many typos and grammar errors that must be addressed
Author Response
Dear reviewer:
On behalf of my coauthors, I would like to re-submit the revised manuscript entitled “Recent advances in peptide inhibitors targeting wild-type Ras protein interactions in cancer therapy” [ijms-3387419] by Qin et al. for your evaluation.
In the past days, we carefully revised the manuscript based on the comments from the editor and all reviewers. We performed the suggested analyses including increasing the citations of references, improving the resolution of images, carefully checking and revising the sentences, introducing subheadings, etc.
We sincerely thank all of you for your in-depth comments/suggestions/questions and careful reading, which helped us tremendously in improving our manuscript. The attachment is the point-to-point answers to the editor’s and reviewers’ comments, and we hope our revisions could satisfy the rigorous requirements of all reviewers and the journal.
Thank you very much for reviewing our manuscript.
Cordially,
Weirong Qin, Ph.D.
Pharmaceutical College,
Guangxi Medical University,
Nanning, China 530021
- mail: qinweirong@sr.gxmu.edu.cn

Round 2
Reviewer 2 Report
Comments and Suggestions for Authors
The authors have substantially improved the manuscript by adding updated references and reorganizing it in a more readable way. All my questions have been properly addressed. I have only a few minor comments:
1) In my opinion, the current paragraph "1. Peptide Inhibitors Targeting Ras Signaling Pathways" is useless and, unless requested by the other reviewer(s), it should be removed and its content included in the final part of the review.
2) Table 1: the authors missed the point of my request, which was essentially to add a column to the table and indicate in that column, for each compound, the literature from which the parameter values in the table were taken. I therefore ask that this addition be made.
3) The last paragraph "Discussion" should be renamed "Conclusions"
I also saw few typos to be corrected.
Author Response
Response to reviewer 2:
The authors have substantially improved the manuscript by adding updated references and reorganizing it in a more readable way. All my questions have been properly addressed. I have only a few minor comments:
Answer: We sincerely thank Reviewer 2 for his/her supportive comments and valuable suggestions.
- In my opinion, the current paragraph "1. Peptide Inhibitors Targeting Ras Signaling Pathways" is useless and, unless requested by the other reviewer(s), it should be removed and its content included in the final part of the review.
Answer: We sincerely thank Reviewer 2 for his/her valuable suggestions.
We have removed "1. Peptide Inhibitors Targeting Ras Signaling Pathways".
We have moved its content to the final part of the review.
- Table 1: the authors missed the point of my request, which was essentially to add a column to the table and indicate in that column, for each compound, the literature from which the parameter values in the table were taken. I therefore ask that this addition be made.
Answer: We sincerely thank Reviewer 2 for his/her valuable suggestions.
We have added a column to the table and have marked the reference in Table 1.
- The last paragraph "Discussion" should be renamed "Conclusions"
Answer: We sincerely thank Reviewer 2 for his/her valuable suggestions and carefully reading.
The last paragraph "Discussion" have been renamed "Conclusions".
I also saw few typos to be corrected.
Answer: We sincerely thank Reviewer 2 for his/her valuable suggestions and carefully reading.
We have corrected some mistakes, such as “Furthermore”, “potentially”, “Ras oncoproteins” and so on.